# Evaluating Machine Learning-Based Classification of Human Locomotor Activities for Exoskeleton Control Using Inertial Measurement Unit and Pressure Insole Data

**DOI:** 10.3390/s25175365

**Published:** 2025-08-29

**Authors:** Tom Wilson, Samuel Wisdish, Josh Osofa, Dominic J. Farris

**Affiliations:** 1Research Software & Analytics Group, University of Exeter, Exeter EX4 4PY, UK; t.wilson6@exeter.ac.uk; 2Public Health & Sport Sciences, University of Exeter Medical School, University of Exeter, Exeter EX4 4PY, UK; s.wisdish@exeter.ac.uk; 3Human Sciences Delivery Team, Defence Science and Technology Laboratory, Salisbury SP4 0JQ, UK

**Keywords:** machine learning, IMU, classification, walking, running, speed

## Abstract

Classifying human locomotor activities from wearable sensor data is an important high-level component of control schemes for many wearable robotic exoskeletons. In this study, we evaluated three machine learning models for classifying activity type (walking, running, jumping), speed, and surface incline using input data from body-worn inertial measurement units (IMUs) and e-textile insole pressure sensors. The IMUs were positioned on segments of the lower limb and pelvis during lab-based data collection from 16 healthy participants (11 men, 5 women), who walked and ran on a treadmill at a range of preset speeds and inclines. Logistic Regression (LR), Random Forest (RF), and Light Gradient-Boosting Machine (LGBM) models were trained, tuned, and scored on a validation data set (*n* = 14), and then evaluated on a test set (*n* = 2). The LGBM model consistently outperformed the other two, predicting activity and speed well, but not incline. Further analysis showed that LGBM performed equally well with data from a limited number of IMUs, and that speed prediction was challenged by inclusion of abnormally fast walking and slow running trials. Gyroscope data was most important to model performance. Overall, LGBM models show promise for implementing locomotor activity prediction from lower-limb-mounted IMU data recorded at different anatomical locations.

## 1. Introduction

Assistive technologies to aid human locomotion have been developed for a range of purposes, from augmenting the user’s physical capability, to guiding gait rehabilitation, to providing everyday assistance with common locomotor tasks. Robotic exoskeletons or exosuits (EXOs) are a specific class of such technology and herein considered wearable devices, usually intended to restore, assist, or augment human locomotion. In the current century, significant advancements have been made in these technologies, leading to adoption in industrial and rehabilitation settings to prevent injury or restore movement [1]. A common goal of EXOs is to reduce the metabolic cost of locomotion through mechanical assistance to the lower limbs, and several such devices have achieved this [2,3,4,5].

Key to the success of powered gait-assist EXOs has been the development of control systems that provide adaptive assistance to accommodate varying gait requirements, such as flat vs. stair climbing (e.g., [4]). To achieve this, a common control architecture employed is a tiered system whose highest level relies on feedback from wearable sensors to classify the wearer’s gait (e.g., walking, running, stair ascent, etc.) [6]. This allows a middle tier to switch to an appropriate control law, which is often implemented at the lowest level by matching measured and desired assistive forces or torques [6]. Successful high-level gait classification for a hip-assist EXO controller has been demonstrated using sensing signals from inertial measurement units (IMUs) mounted on the thigh and shank as inputs to machine learning classifier models [4]. Zhang et al. [4] successfully implemented a boot-strapped random forest classifier model to classify level walking, stair ascent, and stair descent. However, as EXO designs extend their use to a wider range of activities, classification models will likely need to adapt to this new requirement. More broadly, approaches to movement classification may be applicable to a wider range of gait-assist technologies whose control architecture relies on similar classification needs.

One previously identified use case for lower limb EXOs is to assist military personnel during dismounted patrol operations [7,8]. This task is characterised by prolonged periods of patrolling on foot with additional loads, over varied terrain [7]. During prolonged walking, humans adopt a range of speeds. Therefore, speed-adaptive control of walking EXOs has significant benefits [2]. There is also potential for bouts of running during dismounted patrol that an EXO must recognise and accommodate by switching control laws [9]. Furthermore, a variety of other movements will be performed that the EXO is not designed to assist with, and in such cases it must ‘get out of the way’ to avoid hindering the user [7]. Therefore, models that can classify movements an EXO can assist with, movements it cannot assist with, and additional factors such as speed and incline, are of value and should be further explored.

IMUs are practical and low-cost motion sensors with potential for EXO control [10]. In-shoe pressure sensors also provide relevant information for EXO feedback control [10]. However, commercial pressure insoles are expensive and complex, making them less viable options for in-field use. Our team has developed an e-textile-based insole for sensing one-dimensional centre of pressure under the foot. This technology has the potential to provide a simplified and lower-cost solution for generating relevant input for movement classification. Therefore, IMUs and the insole sensors described are candidate technologies for providing inputs to machine learning classifiers for EXO control.

While Zhang et al. [4] employed IMUs on the thigh and shank of the wearer for their hip EXO, other sensor locations may be desirable for alternative EXO designs or for EXOs targeting ankle assistance (e.g., Slade et al. [2] measured ankle kinematics). Locating sensors where hardware already exists is also preferable, as it reduces the number of components and attachment points required in EXO designs. Furthermore, computational and hardware requirements can be minimised by implementing only the necessary number of sensors and signals for gait classification. Therefore, exploring the necessary locations and number of sensors required for effective gait classification will yield useful information for EXO design. Such exploration can be achieved with tree-based classifiers such as the random forest (RF) approach [4]. However, there may also be benefits to using other tree-based methods such as the Light Gradient-Boosting Machine (LGBM) [11] model, which may outperform RF when properly tuned [12].

In this study, we aimed to test the classification of varied human locomotor movements using data from lower body IMU sensors and foot centre-of-pressure sensors. For performance comparison, we implemented three different machine learning classifier models: (1) logistic regression (LR, as a baseline option); (2) random forest (RF); (3) LGBM. Secondly, we aimed to determine the most important and necessary input signals and sensor locations for this purpose. The work described represents an initial exploration of methods for implementation in future EXO designs. The approaches tested may also prove valuable for other gait-assist technologies that rely upon classification of user movement from IMU and instrumented insole sensors.

## 2. Methods

### 2.1. Data Collection

Sixteen healthy volunteers (eleven men and five women; mean ± SD: age 26 ± 5 years, body mass 79 ± 13 kg, stature 178 ± 7 cm) participated in this study. All participants were free from musculoskeletal injury and completed a Physical Activity Readiness Questionnaire (PAR-Q) to confirm their suitability for participation. Prior to initiating any study procedures, each participant provided written informed consent. The study received favourable opinion from the Ministry of Defence Research Ethics Committee (Application No: 2289/MODREC/24) and the Public Health and Sport Sciences Research Ethics Committee at the University of Exeter.

Within a single laboratory visit, each participant was equipped with 7 IMUs (Vicon Blue Trident, VICON, Oxford, UK), affixed to the posterior pelvis and the lateral aspect of both thighs, both shanks, and both feet using hypoallergenic adhesive tape and self-adhesive bandages (Figure 1). These locations were chosen as common attachment points for lower-limb EXO hardware and typical sites used for kinematic modelling of the lower limb. The IMU signals, which included triaxial accelerometer, gyroscope, magnetometer, and orientation data, were synchronously sampled at 225 Hz and acquired using VICON’s Beacon and Lock Lab into Nexus software (Nexus v2.15, VICON, Oxford, UK). Orientations, referred to as ‘Global Angles’, were computed by VICON’s proprietary sensor-fusion algorithm and represented in the default output of exponential coordinates.

Participants also wore custom-made right and left instrumented in-shoe pressure-sensing insoles. Briefly, the insoles comprise an E-textile fabric weave that outputs a voltage corresponding to the anterior–posterior position of the centre of pressure (COP) under the foot. With no or very light pressure exerted by the foot on the insole, the voltage sits at zero. When pressure is exerted, the insole outputs a voltage between 1 and 5 V, dependent on the average location of pressure exerted by the foot (1 V nearest the heel, increasing linearly to 5 V nearest the toes). Example output from multiple walking steps is shown in Figure 2. Participants performed walking and running trials on an instrumented treadmill (M-Gait, Motek, The Netherlands) at a range of speeds and inclines (Table 1). Treadmill locomotion at constant speeds and inclines is mechanically equivalent to overground locomotion at the same speed and incline [13] and was therefore considered an acceptable simulation of real-world mechanics. Data were captured for 30 s once participants acclimatised to each task. Upon completion of a task, participants were allowed adequate rest and mitigate the effects of fatigue before beginning the next task. To provide data for a simple movement task that was not walking or running, participants also completed five separate vertical jumps.

### 2.2. Data Preparation and Preprocessing

To transform the raw IMU data into a format suitable for model training, we first added labels for the activity type (walk, run, or jump), incline, and speed of each trial. We then applied a sliding window aggregation to the data (e.g., Figure 2), extracting various statistics across the window for every signal. The window size and the statistics calculated were treated as variables for the tuning stage (Figure 3). Statistical variables were selected from the following options: minimum, maximum, mean, standard deviation, first, and last. The models then predict the feature values for each window. Overall, the full processed dataset contained 433 columns, due to the combination of signals (Figure 2). This number was substantially reduced during the tuning stage, as including all features was treated as a parameter.

### 2.3. Selection of Machine Learning Models

Three different model types were selected to explore a wider range of approaches, increasing the likelihood of finding a successful predictor and better understanding the strengths and weaknesses of each model and any systemic issues with the data and/or processing. We used a supervised learning task, as the labels targeted for prediction (activity, speed, and incline) are known for the entire dataset. The task was further subdivided into three sub-tasks: predicting activity type is a multiclass classification (one of three discrete classes), whereas predicting speed and incline are regression tasks (predicting a continuous value). These parameters dictated which models were suitable. The three model types chosen were

Logistic/Linear Regression (LR);Random Forest (RF);Light Gradient-Boosting Machine (LGBM).

The LR and RF implementations are available in the scikit-learn Python package v1.7.1 [14], and LGBM is available in the LightGBM Python package v4.6.0 [11]. For each model type, both a classification (activity category prediction) and a regression (speed and incline prediction) algorithm were used. LR is relatively simple to implement and quick to train, but it typically shows limited effectiveness on high-dimensional data compared to more complex models, as it assumes linear relationships. It was therefore selected mainly as an interpretable baseline for comparison with more complex models. RF is a tree-based ensemble learning method, more robust than a single decision tree model. It has been employed successfully in previous work [4] and is a popular choice for both classification and regression tasks. RF has several benefits over LR models; for instance, it can handle non-linear relationships and feature interactions and is less prone to overfitting. However, it can be slower to train and more computationally intensive for large datasets [15].

The LGBM model is also a tree-based ensemble learning method but uses a different tree-building strategy. It can perform similarly to RF models while requiring less intensive training and often outperforming RF when properly tuned. However, due to the number of tuning parameters, implementation can be more complex [12]. These tree ensemble models (RF and LGBM) are typically strongest with medium-sized tabular data such as ours [16], and can capture nonlinear feature interactions without extensive feature engineering. Notably, the relative importance of the data features in the model weightings can be assessed, enabling the identification of, for example, which IMU locations are most important for high-performing models. In turn, this allows us to better understand how the models work, as well as remove extraneous features from training for both computational and performance gains. Models such as Support Vector Machines (SVMs) and K-Nearest Neighbour (k-NN) were also considered. Whilst appropriate for the dataset size, they were ultimately not selected, as both can be computationally expensive and do not provide native feature-importance outputs, unlike RF and LGBM models. In addition, k-NNs typically struggle with class imbalances, as in our activity types [17].

### 2.4. Computing Resources

Data processing and model training was conducted on a virtualised resource on the University of Exeter’s Isca HPC. This configuration included 128 GB vRAM memory and 500 GB storage, running on an Intel Xeon Processor (Cascade Lake). The system was configured with 8 logical CPUs, each presented as a dedicated socket with one core per socket (no simultaneous multithreading enabled). The operating system was running Ubuntu 24.04.1 LTS with kernel version 6.8.0-51-generic.

### 2.5. Models and Experiment Design/Setup

Data from 14 participants were used for training and validating the models, while the data from the remaining 2 participants were reserved as the test set. Once the models had been tuned, trained, and scored on the validation set, they were then scored separately against the test set. This score provides an indication of how generalised the model is—how well it should perform on unseen data. A model that performs well on the validation set may still perform poorly on the test set if it has been overfitted to the training data. We deliberately reserved these two subjects to provide a deployment-like, untouched test set. We acknowledge that Leave-One-Subject-Out cross-validation (LOSOCV) is a complementary method that more directly estimates per-subject generalisability; however, with only 16 participants, LOSOCV estimates can have high variance and substantially increase complexity and computational demand. We therefore regard LOSOCV as a valuable but computationally and statistically noisy alternative, and note it as future work to further characterise between-subject variability.

In each experiment, the data were processed once, after which each model (for activity, speed, and incline prediction) was trained sequentially (Figure 3). Since several of the data processing choices (such as the size of the sliding window aggregation and the proportion of training and validation data) were treated as parameters to be tuned in addition to the model parameters, the models were generally tuned as a group rather than separately. This approach was taken to allow better comparison between the models and to more closely reflect how these models may be used in the future, where new, unseen data would be processed once and then all predictive models applied.

Typically, data were shuffled and split into *n*% train set, (100 − *n*)% validation sets, with *n* between 60% and 90%. Contamination between sets was avoided, and feature distributions were across sets. Activity type data were imbalanced (‘jump’ was severely underrepresented). Speed was largely balanced, although higher speeds were less common, and more extreme inclines were similarly underrepresented. In order to counter this, the splitting algorithm stratified the split to ensure that values of all three features were proportionally balanced across the sets. A warning was raised if the difference exceeded 5%. As this is time-series data, the algorithm also split sequentially rather than shuffling, and did not share participant trial data across sets, to avoid contamination.

When tuning the model parameters, a random search cross-validation method was used. For each parameter, a value range was defined, and then random combinations of values were selected a specified number of times. For each combination, the model was trained and cross-validated (with train and validation sets split differently to avoid overfitting and improve robustness). The highest-scoring combination of parameters (Appendix A) was then used in all subsequent experiments. The models were also tuned by removing irrelevant features: by analysing the model feature importance in initial experiments and iteratively testing reduced feature combinations, a best combination was identified (Figure 2), reducing the total number of data columns from 433 to 65. Whilst the main benefit of this was reduced computational cost, removing irrelevant signals also improved overall model performance.

To address the imbalance in activity types, class weights were computed from the training set and applied as the sample-weight argument during model fitting. No oversampling or synthetic-sample methods (e.g., SMOTE) were used. All weight computation and application were performed only on the training portion to avoid information leakage; validation and test sets remained untouched. The classification results (Figure 4) include per-class scores to make minority-class performance transparent.

The aim of the main experiments was to achieve the highest validation and test scores for predicting activity type, speed, and incline. In addition, experiments were also run using signals from limited numbers of IMUs: pelvis (1) IMU, thighs (2), pelvis and thighs (3), feet (2), and pelvis and feet (3). The purpose of these additional experiments was to help determine which sensors and signals may not be necessary for accurate activity prediction, and therefore not required in a system of sensors used to classify movement types. The experiments with just pelvis and/or thighs were selected because these IMUs appeared to be the most important in previous experiments (see Results). Minor variations on the main experiments were also run to train for activity predict only, in order to explore differences in performance when not balanced with other feature prediction.

### 2.6. Model Evaluation

The performance of each model was evaluated using multiple metrics. For classification, the primary score used was the F_1_ score, a balanced combination of precision and recall. Precision and recall both measure the model’s ability to predict classes correctly, but precision is more sensitive to false positives, whereas recall is more sensitive to false negatives. Since both were of equal interest, the F_1_ score was a suitable metric. A weighted variant of the F_1_ score was also used, which accounts for the class ratios in the data when calculating the overall average score. For both scores, the value can range from 0 to 1, with 1 indicating perfect prediction.

Confusion matrices were also employed, providing a more detailed breakdown of the model’s performance in predicting each class correctly. For regression, the primary score used was R-squared (*R*^2^). For each model type, an average of the classification F_1_ score and the two regression *R*^2^ scores was taken as a general comparative indicator of performance.

In the regression experiments, the speed and incline values were technically discrete in the trial data (since they were set values in each trial). However, the models treated (and predicted) speed and incline as continuous, to better match the physical reality of kinematic data and potential use outside of a laboratory setting. Therefore, some discrepancy was expected between the true and predicted values for speed and incline. The evaluation metrics used accounted for this well, as they weight relative error and variance. Results were plotted as histograms, with the continuous predicted values binned to match the discrete values for easier comparison.

## 3. Results and Discussion

The evaluation scores for the main experiment runs are displayed in Table 2. For the activity classification task, the LGBM model scored the highest with an F_1_ score of 0.87. The LR model achieved a score of 0.80, and the RF model scored the lowest with 0.71. The confusion matrices (Figure 4) suggest that the LGBM and LR models were both highly successful in categorising ‘walk’ and ‘jump’ activities, but tended to misclassify ‘run’ activities as ‘walk’. The RF model suffered from this most of all, as well as a tendency to misclassify ‘jump’ activities as ‘run’. The test scores suggest that the LGBM model is well-generalised, with a score of 0.94. The LR model has a much lower score of 0.53, whereas the RF model had a score of 0.78.

For the speed regression task, the models all performed similarly well, with the RF model achieving the highest score. The histogram plots in Figure 4 show that whilst the prediction distribution (yellow) of the models differs somewhat (RF and LGBM appear more skewed towards lower values, whereas LR is more centred around 1.5), the binned predictions consistently agree with the true values. The high test scores for RF and LGBM suggest that the models are well-generalised (Table 2). The extremely low test score for LR (Table 2) suggests that it overfitted to the training data and completely failed to capture the underlying behaviour.

In general, it appears that incline was more difficult to predict than speed. The LGBM model had the best results with an *R*^2^ score of 0.69, RMSE of 4.1°, and a test *R*^2^ score of 0.62. The histogram plot in Figure 4 suggests that, while the model was moderately successful, it failed to predict more extreme inclines. Overall, the LGBM model performed the best, with an average score of 0.80 for the validation set and 0.84 for the test set across the three tasks. The LR and RF models had similar average validation set scores of 0.70 and 0.71, respectively, but the LR model clearly failed when tested against the test set, whereas the RF model achieved an average score of 0.73.

To investigate the issue of models misclassifying ‘run’ activities as ‘walk’, we repeated the main experiment several more times, discounting files containing a different treadmill speed in each cycle. The rationale for this was that fast-walking speeds and slow running speeds (close to 2 m·s^−1^) are not commonly used unless transitioning between gaits, and rely on mechanics that are less characteristic of walking or running, respectively [18]. Table 3 displays the percentage of correctly identified ‘run’ labels (i.e., top-left corner of the confusion matrices) for each cycle (the ‘None’ row corresponds to the original main experiment results). These results show a marked improvement when the data containing either 1.7 m·s^−1^, 2.2 m·s^−1^, or both, were excluded. These correspond to the fastest walking speed (1.7 m·s^−1^) or slowest running speed (2.2 m·s^−1^), suggesting a potential biomechanical origin to this issue. At these ‘borderline’ speeds, the difference between walking and running gaits may become blurred and individualised. Differences in stature and leg length, for example, might make these speeds suited to walking or running for different individuals (i.e., a long-legged individual might be more able to maintain walking at 1.7 m·s^−1^ vs. a shorter-legged individual). Anatomically normalised metrics of speed, such as Froude number, might therefore be more valuable as labels than absolute speeds. The models reacted slightly differently to the exclusion of selected speeds. RF performed best without 2.2 m·s^−1^, whereas LR and LGBM performed best without 1.7 m·s^−1^. This is likely due to specific model implementations and decision boundaries, but the overall behaviour reflects the inherent soft boundary between walking and running activities. The ‘walk’ classification is likely less susceptible to this issue as it had more training data across a wider range of conditions, such as incline.

The models tuned to classify activity only had better results, as expected. The main difference in the feature selection was the inclusion of both the *y* and *z* dimensions, which typically improved performance for activity classification but reduced incline prediction performance. LGBM scored 0.92, RF scored 0.90, and LR scored 0.87 for the validation set. The test scores were 0.94, 0.94, and 0.56, respectively. This indicates that an even higher score can be achieved when tuned to a single task only, but that a similar pattern in model performance remains, such as the LR model’s tendency to overfit the validation set.

### Feature Importance

As noted in Figure 2, many unimportant features were removed from the dataset during the tuning process. Preliminary analysis at this stage showed that the minimum and maximum statistics were more important than any others, and that whilst the three dimensions were similarly important (with *z* slightly more so), reducing the dimensions from three to one had a negligible impact on scores. We then repeated this feature importance analysis on the trained RF and LGBM models, the results of which are in Appendix A. The gyroscopic outputs were the most important signal, with a mixed ordering of other measures. When considering the average performance across activity, speed, and incline, the general trend in IMU location importance was that anatomically proximal positions such as pelvis and thighs were most important, with the foot sensor signals being least so (Appendix A).

To ascertain which IMUs or combinations of IMUs might be necessary for the models to perform well, we ran experiments with data from only specific IMU locations. The validation and test scores are displayed in Table 4. The overall picture is that these scores are not markedly lower than those from the model trained on the full IMU dataset. The difference between the full model and the highest-scoring model (thighs only) is only 0.04 for the validation set and 0.09 for the test set, yet all of the models were largely successful in the three tasks. They also displayed similar behaviours to the full models in that they typically achieved consistently high scores for activity classification, and for speed prediction to a slightly lesser degree, but struggled more with incline prediction (Table 4).

## 4. Limitations

The present work explored the use of three machine learning models. The choices were rationalised (see Section 1 and Section 2), but it is possible that other models (e.g., neural networks) might provide viable alternatives. The fact that tree-based classification has been implemented in EXO control systems adds weight to the choices made here. However, our study was developmental in nature and did not include any attempt to implement the classification models in real time or as part of an EXO system. The latter issue makes it difficult to contextualise the machine learning models’ performance because we cannot establish thresholds for the F_1_ or R^2^ scores required for successful function of an EXO. Therefore, the work requires further progression before any implementation in a real EXO system. Comparison with more complex models could confirm the selection of LGBM as the most suitable and promising option. This might best be achieved with a larger dataset, including data collected outside the lab with natural, spontaneous changes of locomotor speed and incline. Real-time implementation of movement classification will also be necessary, requiring data transfer from the sensors to onboard (person) computing hardware and software for processing signals and running classification tasks. The present approach performed all processing tasks before passing extracted features to the machine learning models. Therefore, it did not provide a sense of the inference time that is critical for real-time implementation. However, confidence can be drawn from prior real-time implementation of similar models. Furthermore, while IMU attachment locations were chosen to represent likely points of attachment of EXO hardware, testing with data from IMUs connected to an EXO rather than directly to the person will be necessary. Similarly, should technology developers wish to apply the described classification methods to other instrumented gait-assist technologies, they will need to explore the impact of IMUs being placed on the devices themselves.

## 5. Conclusions

In this study we first sought to evaluate selected popular machine learning models for classifying human locomotor activity in terms of gait, speed, and incline. This was motivated by the development of high-level control algorithms for EXOs. Of the evaluated models, the LGBM performed best overall showing good promise for classification of locomotor activity and speed, but not incline. Larger data sets or alternative models (e.g., neural networks) may provide better performance for incline classification. Activity prediction was somewhat clouded by intermediate walking and running speeds, suggesting the models are reliant on features of a biomechanical origin. Of particular interest to EXO designers is that equivalent performance was obtained from very limited subsets of IMU sensors and signals. This suggests that minimal hardware is required and that different locations can be used. Therefore, flexibility exists in choosing where to place sensors and how many to use for this type of classification task. The work provides a platform for future studies to explore EXO-specific IMU configurations, the use of larger real-world (outside the lab) IMU datasets, and other machine learning models that might predict incline with greater success.

## Figures and Tables

**Figure 1 sensors-25-05365-f001:**
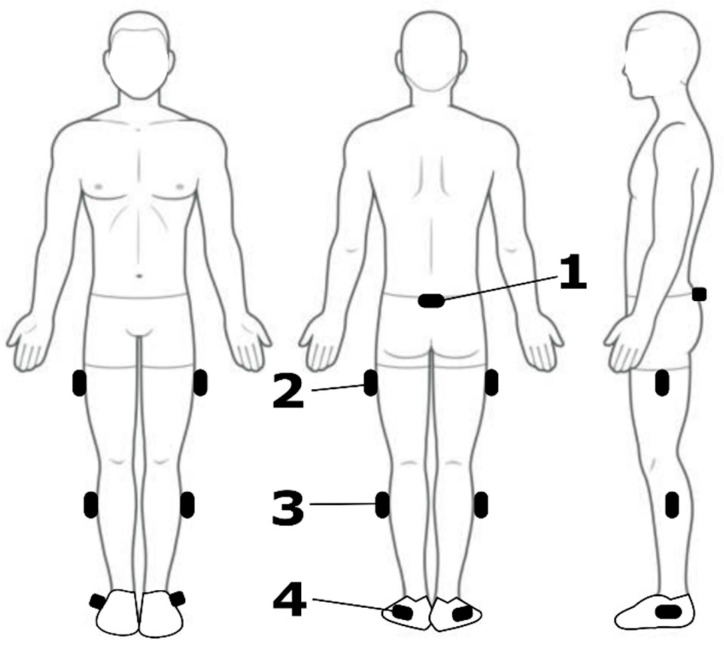
Locations of the IMU sensors on the participant (black rounded rectangles). (1) Posterior pelvis—common attachment of an EXO waist unit containing the drive system and/or computing hardware; (2) lateral thigh—likely attachment site for hip or knee EXOs; (3) lateral shank—suitable attachment site for knee or ankle EXOs; (4) lateral shoe (foot)—ankle EXOs commonly attach to footwear or have a foot attachment section, similar to an ankle–foot orthosis.

**Figure 2 sensors-25-05365-f002:**
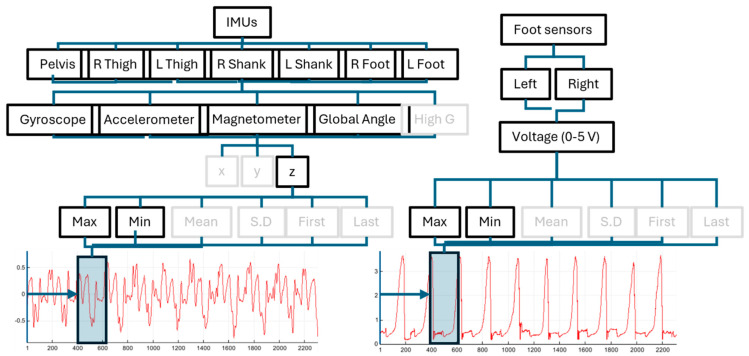
Schematic of signals fed into the machine learning models. Seven IMUs each output five signal types, each with three (x, y, z) components. The foot sensors each output a single voltage channel (e.g., **bottom**, **right**). For each component of each signal, the maximum (max), mean, standard deviation (SD), first, and last values are computed over a sliding window (opaque boxes, **bottom**, **left** and **right**). Following model tuning, the greyed-out signals, components, and/or statistics were excluded.

**Figure 3 sensors-25-05365-f003:**
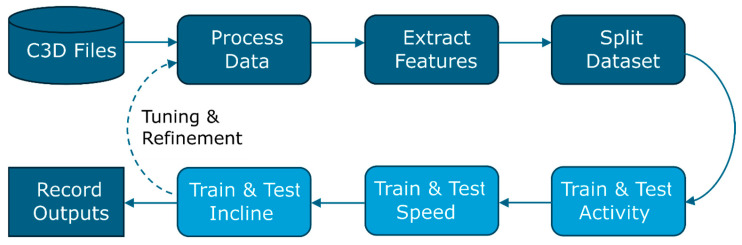
Workflow diagram for the experiments.

**Figure 4 sensors-25-05365-f004:**
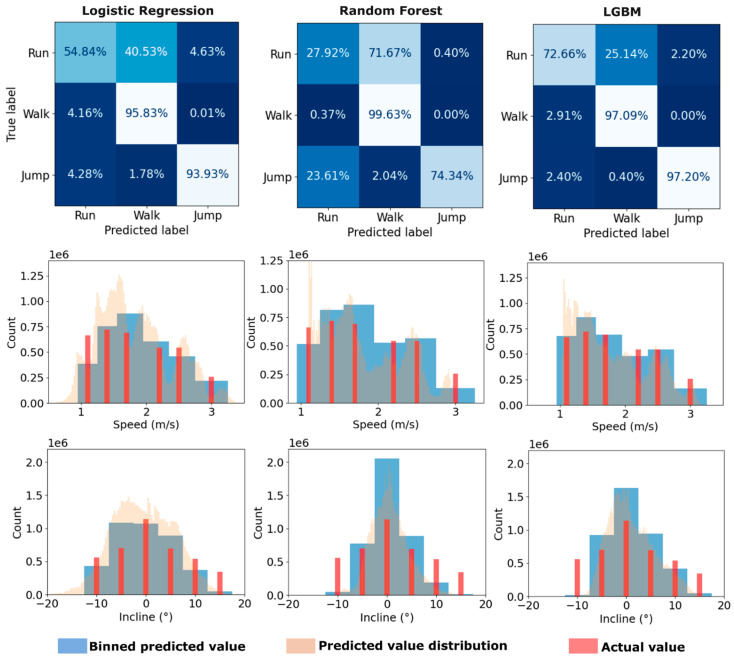
Model performance against the validation set. The first row shows the confusion matrices for the activity classification task. The second and third rows show histograms of the speed and incline predictions, respectively. Axes labelled with 1e6 refer to scientific notation 1 × 10^6^.

**Table 1 sensors-25-05365-t001:** List of activities and conditions completed during the experimental protocol. Only the check-marked combinations of speed and gradient were performed.

Task	Speed (m·s^−1^)	Treadmill Gradient (%)					
		−10	−5	0	5	10	15
Walking	1.1	☑	☑	☑	☑	☑	☑
	1.4	☑	☑	☑	☑	☑	☑
	1.7	☑	☑	☑	☑	☑	☑
Running	2.2	☑	☑	☑	☑	☑	
	2.5	☑	☑	☑	☑	☑	
	3		☑	☑	☑		
Vertical Jump	0			☑			

Blank boxes indicate the task was not performed at the corresponding experimental conditions.

**Table 2 sensors-25-05365-t002:** F_1_ (activity) and R^2^ (speed and incline) scores for all models against the validation and test sets. Values marked with * have been clipped to 0 from an extreme negative value, indicating model failure.

Set	Task	LR	RF	LGBM
Validation	Activity	0.8	0.71	0.87
	Speed	0.81	0.87	0.84
	Incline	0.49	0.53	0.69
	Average	0.7	0.71	0.8
Test	Activity	0.53	0.78	0.94
	Speed	0 *	0.98	0.95
	Incline	0 *	0.44	0.62
	Average	0.18	0.73	0.84

Colours highlight better performance in a scale from red to amber to green (worst to best).

**Table 3 sensors-25-05365-t003:** Percentage of ‘run’ activities correctly classified with different speed trials removed from the data.

Speed Removed (m/s)	‘Run’ Classification Score (%)		
	LR	RF	LGBM
None	55	27	73
1.1	53	33	65
1.4	56	29	67
1.7	61	29	85
1.7 & 2.2	63	57	88
2.2	57	64	75
2.5	50	27	67
3	51	33	63

Shading indicates better scores in dark green, fading to clear background for worse scores.

**Table 4 sensors-25-05365-t004:** F_1_ (activity) and R^2^ (speed and incline) scores for the trained LGBM model with limited IMU configurations against the validation and test sets.

Set	Task	Pelvis	Thighs	Pelvis & Thighs	Feet	Pelvis & Feet
Validation	Activity	0.86	0.90	0.87	0.83	0.90
	Speed	0.81	0.81	0.82	0.74	0.86
	Incline	0.077	0.58	0.56	0.44	0.43
	Average	0.58	0.76	0.75	0.67	0.73
Test	Activity	0.79	0.90	0.91	0.92	0.92
	Speed	0.56	0.87	0.90	0.91	0.92
	Incline	0.22	0.48	0.40	0.13	0.33
	Average	0.52	0.75	0.74	0.65	0.72

Colours highlight better performance in a scale from red to amber to green (worst to best).

## Data Availability

The research data supporting this publication are openly available from the following https://doi.org/10.6084/m9.figshare.29598827. Code for building and running the models tested are available from https://github.com/UniExeterRSE/LISA, accessed on 20 August 2025.

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
