# Peer review of "Evaluating Machine Learning-Based Classification of Human Locomotor Activities for Exoskeleton Control Using Inertial Measurement Unit and Pressure Insole Data"

_sensors, 2025, doi:10.3390/s25175365_

Round 1
Reviewer 1 Report
Comments and Suggestions for Authors
This work presents an evaluation of three machine learning models (Logistic Regression (LR), Random Forest (RF), and Light Gradient-Boosting Machine (LGBM)) to classify human locomotor activities, speed, and incline, based on data collected from inertial measurement units (IMUs) and e-textile insole sensors. The study aims to support the design of high-level control systems for wearable robotic exoskeletons.
The approach is technically sound, and the methodology is clearly presented. The experimental design, use of multiple models, and performance comparisons are appropriate and well-justified. However, I would like to raise some points and suggestions for the authors to consider:
- Although the study focuses on the control of an exoskeleton, it could be enriching for the article (especially the introduction) if a wider number of wearable or assistive technologies in which activity classification is performed from inertial and force sensors will be analyzed. For example, sensors embedded in assistive devices such as crutches, canes, or smart footwear, which perform classifications of everyday activities in the real world. This expanded context could increase the relevance and impact of the work, and could help position the work as part of a broader landscape of assistive mobility and rehabilitation technologies.
- The dataset is relatively small (16 participants, with 2 in the test set). Could the authors comment on how they expect the performance to scale with more diverse populations? Is there any risk of overfitting to this specific sample?
- The paper provides a textual description of the IMU placement, it would significantly improve clarity to include a schematic image or figure showing the anatomical positions of the IMUs on the body.
- The system uses 7 IMUs and e-textile insoles. While a subset of sensors was tested, it remains unclear whether the placement and number of sensors can be realistically used in real-world EXO systems. Are these sensor placements aligned with typical EXO mechanical attachment points?
- The paper focuses on LR, RF and LGBM Machine Learning models. However, only a small mention of other alternatives is made in the discussion. It might strengthen the methodology to clarify whether other models (e.g., SVM, k-NN, etc.) have been explored and/or considered and, if so, why they were discarded. Is this choice motivated by interpretability, computational cost, dataset size, or prior domain expertise? This context would help to more clearly justify the final model selection.
- Although the manuscript explains that the hyperparameter fitting was performed by random search cross-validation, it might be appropriate to report the final values of the hyperparameters used for each model (preferably those with the best results). Presentation of this information in a summary table would facilitate interpretation and comparison of the models.
- It would improve the readability and flow of the manuscript if tables were placed closer to the corresponding sections, where they are first referenced and interpreted. In addition, it is recommended that tables be adjusted to fit the full width of the text column, ensuring that all data are clearly visible and avoiding unnecessary wrapping or formatting problems.
This reviewer recommends acceptance of the work if the authors comply with the recommendations made.
Author Response
Comment 1 - Although the study focuses on the control of an exoskeleton, it could be enriching for the article (especially the introduction) if a wider number of wearable or assistive technologies in which activity classification is performed from inertial and force sensors will be analyzed. For example, sensors embedded in assistive devices such as crutches, canes, or smart footwear, which perform classifications of everyday activities in the real world. This expanded context could increase the relevance and impact of the work, and could help position the work as part of a broader landscape of assistive mobility and rehabilitation technologies.
Response 1 - We appreciate the reviewer’s suggestion to broaden the scope of impact here. We had originally focussed on the application to Exoskeletons because the IMUs were placed on the body and because the classification task described is more appropriate to the specific use case of more active military personnel than those using rehabilitative technologies with embedded sensors. However, the general approach may be applicable to other devices and use cases, so we have added statements through the introduction and in the limitations part of the discussion to cover this (highlighted in revised manuscript).
Comment 1 - The dataset is relatively small (16 participants, with 2 in the test set). Could the authors comment on how they expect the performance to scale with more diverse populations? Is there any risk of overfitting to this specific sample?
Response 2 - We recognise that the sample size of 16 participants limits statistical power and the ability to detect modest differences in model performance across participants. The primary aim of the present work was exploratory - to investigate feasibility and trends in model performance for human activity classification with wearable sensors - rather than to test a specific hypothesis at a predetermined power level. While the sample size is consistent with other early-stage wearable sensor studies, we acknowledge that larger and more diverse participant cohorts will be required to provide high-precision estimates of performance and robust assessments of generalisability. We have added a statement to the Limitations sections clarifying this point. We also now note more clearly in the introduction that the work is somewhat exploratory in nature.
Comment 3 - The paper provides a textual description of the IMU placement, it would significantly improve clarity to include a schematic image or figure showing the anatomical positions of the IMUs on the body.
Response 3 - A new figure has been added (now figure 1) that shows the locations of the IMU sensors
Comment 4- The system uses 7 IMUs and e-textile insoles. While a subset of sensors was tested, it remains unclear whether the placement and number of sensors can be realistically used in real-world EXO systems. Are these sensor placements aligned with typical EXO mechanical attachment points?
Response 4 -The placements of the sensors were specifically chosen as likely points of attachment for a lower limb exoskeleton. As noted above, the sensor locations are now included in a figure (Figure 1) and this figure caption also describes how these correspond to likely attachment points for an assistive EXO.
Comment 5 - The paper focuses on LR, RF and LGBM Machine Learning models. However, only a small mention of other alternatives is made in the discussion. It might strengthen the methodology to clarify whether other models (e.g., SVM, k-NN, etc.) have been explored and/or considered and, if so, why they were discarded. Is this choice motivated by interpretability, computational cost, dataset size, or prior domain expertise? This context would help to more clearly justify the final model selection.
Response 5 - This is a good point, and the initial research we conducted into alternate models is currently under-represented in the paper. We have added a paragraph to the Methods section (Selection of machine learning models) discussing some alternate models, including further explanation as to their strengths, and why they were ultimately deemed to be unsuitable.
Comment 6 - Although the manuscript explains that the hyperparameter fitting was performed by random search cross-validation, it might be appropriate to report the final values of the hyperparameters used for each model (preferably those with the best results). Presentation of this information in a summary table would facilitate interpretation and comparison of the models.
Response 6 - We have added a table to the supplementary section detailing the hyperparameter values for the tuned models, with a reference in the main text. We have also indicated that these values, along with the trained models, are publicly available in the code repository.
Comment 7 - It would improve the readability and flow of the manuscript if tables were placed closer to the corresponding sections, where they are first referenced and interpreted. In addition, it is recommended that tables be adjusted to fit the full width of the text column, ensuring that all data are clearly visible and avoiding unnecessary wrapping or formatting problems.
Response 7 - We are not sure what the reviewer’s version of the manuscript looks like, but in the submitted word document, the tables are within 1-2 paragraphs of the first mention, which seems appropriate to us. Presumably the positioning in the final manuscript will be dependent on the proofing and typesetting process.
Comment 8 - This reviewer recommends acceptance of the work if the authors comply with the recommendations made.
Response 8 - We appreciate the reviewer’s recommendation and helpful comments. We hope that we have addressed them sufficiently.
Reviewer 2 Report
Comments and Suggestions for Authors
This study addresses a highly relevant and timely topic with potential to meaningfully advance the field of robotic exoskeletons. The manuscript is generally well-structured and clearly written. I commend the authors for the quality of the work.
That said, I would like to raise a few minor concerns, primarily related to statistical methodology. Addressing these points—particularly by further elaborating and justifying specific decisions—would strengthen the manuscript and enhance its scientific rigor. I hope the following comments are received in the spirit of constructive feedback, aimed at supporting the further development of this valuable contribution.
Sample Size and Statistical Power
The current study does not report whether an a priori power analysis was conducted to determine the required sample size for achieving sufficient statistical power. Although the use of 16 participants may be acceptable for exploratory work with wearable sensors, the absence of a formal power calculation raises concerns regarding the robustness and generalisability of the findings. Without such analysis, it is difficult to assess whether the sample is adequately powered to detect meaningful differences or model performance variations across conditions or participants.
Model Selection and Justification
While the study employs Logistic Regression (LR), Random Forest (RF), and Light Gradient-Boosting Machine (LGBM) models, the rationale behind the inclusion of LR and RF could be further developed. Logistic Regression, although useful as a baseline, is known to underperform with high-dimensional, non-linear data such as IMU time-series signals. Similarly, Random Forest, while a popular and interpretable model, may not be optimal compared to modern alternatives such as deep learning architectures, which have shown improved performance in similar locomotion classification tasks. A clearer discussion on why these models were selected over others—particularly in the context of real-time exoskeleton control—would enhance the study’s methodological clarity.
Statistical Methods
Although the authors employ random search cross-validation from two participants for final testing, the study does not implement Leave-One-Subject-Out Cross-Validation (LOSOCV). This omission limits the ability to evaluate a generalisability of the models to unseen individuals—a critical requirement for human-centered wearable systems such as exoskeletons.
Author Response
Sample Size and Statistical Power
Comment 1 - The current study does not report whether an a priori power analysis was conducted to determine the required sample size for achieving sufficient statistical power. Although the use of 16 participants may be acceptable for exploratory work with wearable sensors, the absence of a formal power calculation raises concerns regarding the robustness and generalisability of the findings. Without such analysis, it is difficult to assess whether the sample is adequately powered to detect meaningful differences or model performance variations across conditions or participants.
Response 1 - We appreciate the reviewer’s comment regarding a priori power analysis. This study did not include a formal a priori calculation, and we recognise that the sample size of 16 participants limits statistical power and the ability to detect modest differences in model performance across participants. The primary aim of the present work was exploratory - to investigate feasibility and trends in model performance for human activity classification with wearable sensors - rather than to test a specific hypothesis at a predetermined power level. While the sample size is consistent with other early-stage wearable sensor studies, we acknowledge that larger and more diverse participant cohorts will be required to provide high-precision estimates of performance and robust assessments of generalisability. We have added a statement to the Limitations sections clarifying this point. We also now note more clearly in the introduction that the work is somewhat exploratory in nature.
Model Selection and Justification
Comment 2 - While the study employs Logistic Regression (LR), Random Forest (RF), and Light Gradient-Boosting Machine (LGBM) models, the rationale behind the inclusion of LR and RF could be further developed. Logistic Regression, although useful as a baseline, is known to underperform with high-dimensional, non-linear data such as IMU time-series signals. Similarly, Random Forest, while a popular and interpretable model, may not be optimal compared to modern alternatives such as deep learning architectures, which have shown improved performance in similar locomotion classification tasks. A clearer discussion on why these models were selected over others—particularly in the context of real-time exoskeleton control—would enhance the study’s methodological clarity.
Response 2 - We have further augmented the discussion of the selected models in the Methods section , including a comparison with alternate models such as Support Vector Machines and k-Nearest Neighbours.
Statistical Methods
Comment 3 - Although the authors employ random search cross-validation from two participants for final testing, the study does not implement Leave-One-Subject-Out Cross-Validation (LOSOCV). This omission limits the ability to evaluate a generalisability of the models to unseen individuals—a critical requirement for human-centered wearable systems such as exoskeletons.
Response 3 - Thank you for the suggestion – we have added a paragraph to section 2.5, further clarifying our choice and discussing why LOSOCV was not used here, but acknowledging its suitability for future work.
Reviewer 3 Report
Comments and Suggestions for Authors
This article presents an applied research in the field of assistive exoskeletons and the use of machine learning for human locomotion classification. It employs IMU sensors and textile insoles with models such as Logistic Regression, Random Forest, and LightGBM to classify activity type, speed, and surface incline.
Suggestions for Improvement
- While the goal of locomotion classification is stated, it is not entirely clear whether this is a conceptual proof or if the system is intended for implementation in real hardware. Suggestion: Add a brief section at the beginning or end outlining the next steps toward real-world exoskeleton deployment.
- Although the authors mention that the system was not integrated into an EXO, it would be helpful to know whether the models offer inference times compatible with real-time systems.
- Class imbalance is briefly mentioned e.g., ‘jump’ being underrepresented, but it is unclear whether balancing techniques such as oversampling or SMOTE were applied. Suggestion: Include a section explaining how this imbalance was handled, or if it was assumed during training.
- The dataset is quite limited, with only 16 participants, and only 2 used for testing, which may compromise the robustness and generalizability of the results.
- In some cases, values are presented without units or in inconsistent formats, e.g., “Speed -67”, “Incline -136” in Table 2. This appears to be a numerical artifact from poorly fitting linear regression models, likely due to extrapolation.
Author Response
Comment 1 - While the goal of locomotion classification is stated, it is not entirely clear whether this is a conceptual proof or if the system is intended for implementation in real hardware. Suggestion: Add a brief section at the beginning or end outlining the next steps toward real-world exoskeleton deployment.
Response 1 - This is a helpful suggestion and we have now added some context at the end of the introduction and the limitations section of the discussion. In these sections we note the exploratory nature of the work and what must be done to progress towards implementation in a real EXO system.
Comment 2 - Although the authors mention that the system was not integrated into an EXO, it would be helpful to know whether the models offer inference times compatible with real-time systems.
Response 2 – Although set in the context of EXO control, the testing carried out here was much more towards the early developmental/exploratory end to examine candidate approaches for taking forward into development for a real time setup. The processing of data was run on the full data set and then passed to the classifiers/regression models, so it is not really possible to consider how the approach performed in a real-time sense. In the revised manuscript, we have emphasised the exploratory nature of the work in the introduction and made clear in the limitations that a real time formulation of the approach is a future requirement.
Comment 3 - Class imbalance is briefly mentioned e.g., ‘jump’ being underrepresented, but it is unclear whether balancing techniques such as oversampling or SMOTE were applied. Suggestion: Include a section explaining how this imbalance was handled, or if it was assumed during training.
Response 3 - As suggested, we have added a paragraph to section 2.5 detailing how class imbalance is handled – we have stated that oversampling and SMOTE were not used, but class weights were applied during model fitting.
Comment 4 - The dataset is quite limited, with only 16 participants, and only 2 used for testing, which may compromise the robustness and generalizability of the results.
Response 4 -We recognise that the sample size of 16 participants limits statistical power and the ability to detect modest differences in model performance across participants. The primary aim of the present work was exploratory - to investigate feasibility and trends in model performance for human activity classification with wearable sensors - rather than to test a specific hypothesis at a predetermined power level. While the sample size is consistent with other early-stage wearable sensor studies, we acknowledge that larger and more diverse participant cohorts will be required to provide high-precision estimates of performance and robust assessments of generalisability. We have added a statement to the Limitations sections clarifying this point. We also now note more clearly in the introduction that the work is somewhat exploratory in nature.
Comment 5 - In some cases, values are presented without units or in inconsistent formats, e.g., “Speed -67”, “Incline -136” in Table 2. This appears to be a numerical artifact from poorly fitting linear regression models, likely due to extrapolation.
Response 5 - As the reviewer notes, R squared values are typically in the range of 0-1, so it may be confusing to see negative integers. Whilst these values are technically valid, we have now clipped them to zero, and updated the three-score average to match. We have added text explaining this, highlighted that this indicates model failure. We hope that this communicates the model performance more clearly, and is now more comparable with the other cases.